# GE-PEFT: Gated Expandable Parameter-Efficient Fine-Tuning for Continual Learning

## Abstract

Continual learning (CL) is a research field focused on continuously adapting foundation models such as large language models (LMs) to newly emerging information sources and tasks. While aspects such as parameter efficiency, knowledge transfer, and managing model capacity have recently received attention, the main research focus in CL remains on preventing catastrophic forgetting. Specifically, there is a lack of solutions that address all these aspects simultaneously. We bridge this gap by introducing Gated Expandable Parameter-Efficient Fine-Tuning (GE-PEFT). Our approach shares knowledge of previous tasks through leveraging a single, dynamically expanding PEFT module within LMs while selectively gating irrelevant previous tasks. Our experiments across multiple task-incremental CL benchmarks demonstrate that GE-PEFT outperforms existing state-of-the-art CL approaches in both full CL and few-shot settings. Our ablation and parameter sensitivity studies highlight the benefit of each proposed component, demonstrating that GE-PEFT offers a more efficient and adaptive solution for CL in LMs.

## 1 Introduction

In recent years, pre-trained foundation language models (LMs) trained on vast amounts of textual data have rapidly advanced state-of-the-art performance in a wide range of natural language processing (NLP) tasks due to the knowledge inherent in them (Petroni et al., 2019). As the world continues to evolve, the available knowledge changes with existing information becoming outdated or receiving updates, and new information becoming available. To keep up with continuous progress and allow LMs to face newly emerging problems, the research domain of continual learning (CL) focuses on the continuous adaptation of LMs to new and updated information, as well as newly emerging tasks. Within this growing field, recent works have identified several desirable properties of CL approaches, including the prevention of catastrophic forgetting (Kirkpatrick et al., 2017), parameter efficiency (Omeliyanenko et al., 2023; Wang et al.; Razdaibiedina et al., 2023), and knowledge transfer (Razdaibiedina et al., 2023).

While research exists on each individual criterion for CL, to the best of our knowledge, only one work partially addresses all criteria simultaneously. Wang et al. (2024) extend the work of Razdaibiedina et al. (2023) who integrate a weak knowledge transfer mechanism into existing parameter-efficient fine-tuning (PEFT) strategies for CL, which initializes new and entirely separate PEFT modules for a new task and prepend it with PEFT weights from previous tasks. Wang et al. (2024) further enhance this approach by providing a learned similarity function that identifies relevant prior tasks and reuses PEFT modules only from these tasks for initialization of the current task, thus gating irrelevant tasks through similarity. While this approach achieves state-of-the-art results in CL it only achieves knowledge transfer during initialization of new tasks, leading to the transferred knowledge being overwritten during training, leaving room for further improvements.

In this work, we provide an alternative strategy for obtaining all desirable CL criteria while providing knowledge of previous tasks to the current task throughout the entire training and inference. We propose the integration of a single Gated Expandable PEFT (GE-PEFT) module into the pre-trained LM. Within this module, a gating mechanism is leveraged to prevent catastrophic forgetting of previous tasks. By using one GE-PEFT module to train all tasks, knowledge transfer is seamlessly integrated during training and inference. The gating mechanism tracks the weights already allocated to previous tasks and ensures that only unused weights are available for updates by subsequent tasks,

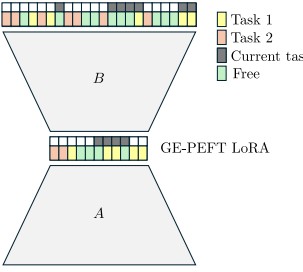 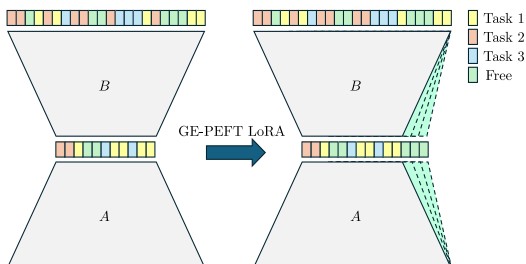

(a) Gating mechanism of GE-PEFT. Task colors indicate neurons reserved for previous tasks. Note that the current task can select both reserved and free neurons. Reserved neurons are only used in the forward pass, while free neurons are updated in the backward pass and reserved for the current task after training.

(b) Expansion mechanism of GE-PEFT. Expanding on LoRA increases the intermediate size through increasing rank, providing more free neurons for future tasks while preserving knowledge from existing tasks.

Figure 1: Visualization of Gated Expandable PEFT (GE-PEFT) on a LoRA PEFT module with gating shown in Figure 1a and growing capabilities shown in Figure 1b.

as illustrated in Figure 1a for a LoRA PEFT module. However, the use of a single module introduces the risk of saturation, i.e., a shortage of unused weights and thus capacity for new tasks. This challenge is addressed by the growing capabilities of our method. We propose a dynamic growth mechanism that allows the model to adjust its size, i.e., the total number of learnable parameters, to meet the requirements of the current task without disrupting previously learned tasks. We illustrate this expansion mechanism on a LoRA PEFT module in Figure 1b.

We demonstrate our GE-PEFT strategy by integrating it into two popular PEFT methods, Low Rank Adapters (LoRA) (Hu et al., 2021), and prefix tuning (Li & Liang, 2021) employed in four popular LMs, BERT (Devlin et al., 2019), AfroXLMR (Alabi et al., 2022), T5 (Raffel et al., 2020) and LLaMA2-7B (GenAI, 2023). Experimental results across multiple task-incremental CL benchmarks in both full CL and few-shot settings demonstrate that GE-PEFT consistently outperforms state-of-the-art methods.

Our contributions are summarized as follows: (1) We propose GE-PEFT, a new model that covers all four desired properties for CL, including knowledge transfer, catastrophic forgetting prevention, parameter efficiency, and scalability. (2) We evaluate GE-PEFT on multiple datasets and small- and large-scale language models. (3) We conduct ablation and parameter sensitivity studies, demonstrating the benefit of each proposed component and the adaptability of the model. (4) We make code and data available to foster reproducibility and further research.[1]

## 2 RELATED WORK

**Continual learning** CL works can be broadly separated into replay-based, regularization-based, and parameter isolation-based Wang et al. (2024) methods. Regularization-based approaches (Kirkpatrick et al., 2017; Aljundi et al., 2018; de Masson D'Autume et al., 2019; Huang et al., 2021; Sun et al.) incorporate regularization terms into the loss function to penalize changes to critical parameters for previously learned tasks. Replay-based approaches (Rebuffi et al., 2017; Chaudhry et al., 2018; 2021; Mirzadeh et al., 2020) retain a subset of training data from previous tasks and use it in conjunction with new data when training on subsequent tasks. However, access to prior task data may not always be feasible and storing it can lead to storage overhead. Additionally, while regularization-based and replay-based methods significantly mitigate the issue of catastrophic forgetting, they do not completely eliminate it. In contrast, our approach leverages gating to fully prevent catastrophic forgetting and does not require additional data to be retained for future training.

Among parameter isolation-based CL, certain architectures use low-rank factorization to separate neural layers into task-specific and shared parameters Hyder et al. (2022). However, this approach requires significant structural modifications to complex LMs. Early works on LMs update the entire

---

[1]Code and data will be openly released on github upon publication: https://shorturl.at/lxidM

LM, e.g., with a growing strategy that enlarges the model for each new task and freezes the previous parameters Rusu et al. (2016) or a gating mechanism that reserves neurons for a specific task Serra et al. (2018). Alternatively, Wortsman et al. (2020) learns a binary gating mask for all model parameters, revealing a highly performing sub-network for each task. All these approaches, however, are not parameter efficient as they require updates to an entire model. Wang et al. (2023) propose a PEFT solution that integrates dedicated parameters into a backbone LM to learn new tasks. While this enables parameter efficient CL, PEFT modules are separated which prevents knowledge transfer. Razdaibiedina et al. (2023) introduce progressive prompts, which extends the prefix tuning PEFT approach with a prepending strategy to share knowledge from previous tasks. Wang et al. (2024) further add a learned similarity-based weighting scheme to this prepending strategy. This enables the gating of irrelevant tasks, but enables knowledge sharing only at training start, as subsequent training overrides the parameters shared at initialization. In contrast to these works, our GE-PEFT enables gated expandable and parameter-efficient CL while providing previous knowledge throughout the entire training and inference process.

**PEFT**    Within PEFT, multiple different strategies have been proposed that integrate a new set of trainable parameters into different parts of the LM while keeping the base parameters frozen to prevent changes. Houlsby et al. (2019) introduce Adapters consisting of multiple layers that are incorporated between intermediate layers of LMs. Hu et al. (2021) propose Low Rank Adapters (LoRA) which provide additive changes to the LM. Li & Liang (2021) introduce prefix tuning where additional weights are added through additional input tokens which are learned during training. These approaches not only enable efficient adaptation of LMs to new knowledge and tasks but also address the problem of catastrophic forgetting of initial model weights. By injecting a dedicated set of new parameters for each new problem, PEFT methods can also prevent the degradation of performance on previously learned tasks. However, this approach prevents any knowledge sharing between tasks.

**Growing networks**    Multiple growing strategies for neural networks have been proposed in related work. Rusu et al. (2016) propose a lifelong learning strategy that enlarges the entire network and copies previous parameters, which results in large numbers of used parameters. Further work focuses on optimizing PEFT module sizes specifically (Zhang et al.; Valipour et al., 2023) but they do not address CL and transfer learning simultaneously.

## 3    METHODOLOGY

### 3.1    PEFT FOUNDATIONS

PEFT modules, as introduced by Houlsby et al. (2019); Hu et al. (2021); Li & Liang (2021), consist of a small set of additional trainable parameters inserted into a pre-trained backbone LM. During training on new data or tasks, the backbone model remains frozen and only the parameters of the inserted PEFT module are updated. This setup allows the model to learn new knowledge or tackle new tasks while remaining parameter-efficient. In our work, we follow the experimental setup of Wang et al. (2024) and thus apply our GE-PEFT architecture to the PEFT models of Hu et al. (2021); Li & Liang (2021).

**LoRA**    The Low Rank Adapter (LoRA) method (Hu et al., 2021) adjusts a pre-trained weight matrix $W_l \in \mathbb{R}^{d \times k}$ of the LM at layer $l$ with input and output dimensions $d$ and $k$. For such a weight matrix, LoRA incorporates a low-rank adaptation PEFT module. For a given task $(t)$, this module consists of two task-specific trainable matrices, $A_l^{(t)} \in \mathbb{R}^{d \times r}$ and $B_l^{(t)} \in \mathbb{R}^{r \times k}$, with rank $r \ll \min(d, k)$. During training, the weight matrix $W_l$ remains frozen, and only the matrices $A_l^{(t)}$ and $B_l^{(t)}$ are updated. For a given input $x_l$ to layer $l$, the input is passed through both $W_l$ and the matrices $A_l^{(t)}$ and $B_l^{(t)}$ and the outputs are combined via element-wise summation, producing the final output $h_l^{(t)}$ as follows:

$$h_l^{(t)} = W_l x_l^{(t)} + B_l^{(t)} A_l^{(t)} x_l^{(t)} \tag{1}$$

While this approach is applicable to all pre-trained layers within the LM, the authors only apply it to all attention layers for additional parameter-efficiency (Hu et al., 2021).

**Prefix tuning**   The prefix tuning method (Li & Liang, 2021) adds $p$ task-specific, trainable, continuous vectors, called the *prefix* $P_p^{(t)} \in \mathbb{R}^d$, to the input $x^{(t)}$ of task $(t)$, which are new tokens with same size as the model's hidden dimensionality $d$. This is achieved using the concatenation operation $[\cdot||\cdot]$, resulting in the adjusted LM input

$$x'^{(t)} = [P_1^{(t)}|\ldots|P_p^{(t)}|x^{(t)}]. \tag{2}$$

## 3.2   GATED PEFT

In PEFT-based solutions, a new PEFT module is inserted into the backbone model to enable training on new data. Each PEFT module operates independently within the model and does not share weights with other modules. As a result, knowledge transfer between PEFT modules is not possible. Inspired by Serra et al. (2018); Ke et al. (2021), we propose to facilitate knowledge transfer using a single PEFT module that is shared across all tasks and datasets. Since only a single PEFT module is inserted into the backbone model, a gating strategy with masking is used to prevent catastrophic forgetting that could result from continuously training the PEFT module on new data.

For each new task or dataset $(t)$, a trainable mask $m^{(t)}$ is computed with the same dimensionality as the activated neurons within the PEFT modules. This binary mask is derived from task embeddings $e^{(t)}$ that take as input a numerical task identifier and transformed into a pseudo-gating function using a Sigmoid activation and a scaling hyperparameter $s$ by

$$m^{(t)} = \sigma(se^{(t)}). \tag{3}$$

The hyperparameter $s$ is selected as a positive scalar that gradually increases in value to $\gg 1$ during training. This forces the learned mask $m^{(t)}$ to converge towards 0 or 1 during training. The resulting pseudo-binary masks $m^{(t)}$ are multiplied element-wise with the respective active neurons within the PEFT modules during training.

**Gated LoRA**   For the LoRA-based solution, the gating mask is applied as

$$h_l'^{(t)} = W_l x_l^{(t)} + (B_l(A_l x_l^{(t)} \otimes m_{A_l}^{(t)}) \otimes m_{B_l}^{(t)}) \tag{4}$$

where $m_{A_l}^{(t)} \in \mathbb{R}^r$ and $m_{B_l}^{(t)} \in \mathbb{R}^k$ are the learned gating masks of task $(t)$ for the LoRA weight matrices $A$ and $B$, respectively. Note that $A$ and $B$ are not conditional to $(t)$, as a single LoRA adapter is shared across all tasks.

**Gated prefix tuning**   For prefix tuning, the gating mask is applied by

$$x''^{(t)} = [P_1 \otimes m_1^{(t)}|\ldots|P_p \otimes m_p^{(t)}|x^{(t)}], \tag{5}$$

where $m_i^{(t)} \in \mathbb{R}^d$ is the learned gating mask of task $(t)$ for the shared prefix $P_i$.

The following steps apply to both gated prefix tuning and gated LoRA. For the former, the steps are applied to each prefix token mask $m_1^{(t)}, \cdots, m_p^{(t)}$, while for the latter, operations are conducted on both masks, $m_{A_l}^{(t)}$ and $m_{B_l}^{(t)}$ per layer $l$, though for simplicity of notation, this distinction is not explicitly made.

Since the pseudo-gating function often produces non-binary masks that can still lead to catastrophic forgetting, we apply binarization of all masks for a task once that task has been fully trained, as suggested by Ke et al. (2022).

$$m_{eval}^{(t)} = \begin{cases} 1 & \text{if } \sigma(se^{(t)}) > 0.5 \\ 0 & \text{otherwise.} \end{cases} \tag{6}$$

This enables tasks to directly access and exclude knowledge from previous tasks. However, weight updates to neurons already used before could lead to catastrophic forgetting of previous tasks $t^{(prev)}$ that have been learned prior to task $(t)$. To prevent this, a mask that aggregates all neurons used in previous tasks $m^{(prev)}$ is calculated through

$$m^{(prev)} = \text{MaxPool}\Big(\big\{m_{eval}^{(t')}, t' \in \{t^{(prev)}\}\big\}\Big). \tag{7}$$

The mask $m^{(prev)}$ is applied to the gradients $g^{(t)}$ to restrict the updates on these neurons during backpropagation while training task $(t)$ through

$$g'^{(t)} = g^{(t)} \otimes (1 - m^{(prev)}). \tag{8}$$

To prevent early exhaustion of the shared PEFT module parameters, Serra et al. (2018) highlight the need for sparsity of the mask $m^{(t)}$. To enforce sparsity, they propose a regularization term to the loss function $\mathcal{L}$ for the currently trained task $(t)$, that we adapt for our architecture. This term regulates the mask values during training based on the number of neurons that are not yet occupied by previously learned data through iterating over all masks with

$$\mathcal{L}' = \mathcal{L} + \lambda \cdot \frac{\|m^{(t)} \otimes (1 - m^{(prev)})\|_1}{\|1 - m^{(prev)}\|_1} \tag{9}$$

where $\lambda$ is a weighting hyperparameter. Note that this regularization, depending on the PEFT technique, is by summation also applied to *all* masks per prefix token or per layer and matrix, as mentioned earlier.

### 3.3 EXPANDABLE PEFT

Using a single gated PEFT module across all tasks enables the model to access the knowledge obtained from previous tasks while preventing catastrophic forgetting. During training on a new task, a portion of the available neurons is reserved and cannot be updated. Consequently, the capacity to integrate new knowledge into the model with this PEFT solution is limited, as the available neurons may become exhausted over time. To address these limitations, we propose an expandable PEFT module that dynamically adjusts its size to meet current needs while still enabling knowledge transfer and mitigating the catastrophic forgetting effect.

**Expandable LoRA** In our proposed solution, given a LoRA module with weights $A \in \mathbb{R}^{d \times r}$, $B \in \mathbb{R}^{r \times k}$, we dynamically introduce additional neurons with their parameters $A_{exp} \in \mathbb{R}^{d \times r_{exp}}$, $B_{exp} \in \mathbb{R}^{r_{exp} \times k}$ by

$$A' = [A^\top | A_{exp}^\top]^\top, \quad B' = [B | B_{exp}], \tag{10}$$

where $[\cdot|\cdot]$ denotes the concatenation operation along the first dimension, $r$ represents the current LoRA rank, and $r_{exp}$ denotes the number of neurons added to expand the layer size. Note that we only extend the intermediate dimension $r$, as both $d$ and $k$ are fixed by the frozen LM. When using our proposed gating approach in conjunction with expandable LoRA, the dimensionality of all masks $m^{(t)}$ is also adjusted by extending the dimensionality of the task embedding $e^{(t)}$ with $e_{exp}^{(t)} \in \mathbb{R}^{r_{exp}}$ through

$$e'^{(t)} = [e^{(t)} | e_{exp}^{(t)}]. \tag{11}$$

**Expandable prefix tuning** For prefix tuning with prefix $P_i \in \mathbb{R}^d$ we dynamically introduce $p_{exp}$ additional prefix vectors $P_{exp} \in \mathbb{R}^d$ to the model input with

$$x'^{(t)} = [P_1^{(t)} | \ldots | P_{p+p_{exp}}^{(t)} | x^{(t)}], \tag{12}$$

where $p$ represents the current prefix length and $p_{exp}$ denotes the number of prefix vectors added to expand the prefix size. When used in conjunction with gated PEFT, all added prefix vectors are additionally multiplied by new masks $m_{p+j}^{(t)} \in \mathbb{R}^d$ for all added shared prefixes $j \in [1, \ldots, p_{exp}]$.

**When to Expand PEFT** While several strategies of varying complexity exist for determining when to expand neural networks, such as predefined schedules (Evci et al., 2022), we adopt a simpler approach, using an established strategy that expands the network when the loss reaches a plateau (Wu et al., 2019; Kilcher et al., 2018). To be precise, once the validation loss reaches a plateau, we perform a PEFT expansion step and resume training of the same task with additional trainable parameters. This process is repeated until no further improvement in task validation performance is observed, even after adding more parameters. The validation loss is calculated at the end of each epoch and early stopping is applied with a patience of 5 epochs, following Wang et al. (2024).

# 4 EXPERIMENTAL SETUP

## 4.1 DATASETS

Following Wang et al. (2023); Razdaibiedina et al. (2023), we utilize the near-domain benchmarks AfriSenti and Web-of-Science (WOS) (Kowsari et al., 2016), as well as the far-domain benchmark MTL5 (Zhang et al., 2015), for our experiments. The near-domain benchmark consists of closely related tasks. WOS is a document classification dataset with a hierarchical structure, consisting of 7 parent classes (biochemistry, civil engineering, computer science, electrical engineering, medical science, mechanical engineering and psychology), each with 5 closely related child subclasses. In line with Wang et al. (2024), we structure our experiments into 7 CL tasks based on these high-level parent classes. AfriSenti (Muhammad et al., 2023) is a multilingual sentiment analysis dataset comprising 12 African low-resource languages (Algerian Arabic (dz), Amharic (am), Hausa (ha), Igbo (ig), Kinyarwanda (kr), Moroccan Arabic (ma), Mozambican Portuguese (pt), Nigerian Pidgin (pcm), Swahili (sw), Twi (twi), Xitsonga (ts), and Yoruba (yo)). Following Wang et al. (2024), we conduct our experiments using three different task orders from the AfriSenti dataset. MTL5 is a widely used far-domain CL benchmark comprising five distinct text classification tasks. AG News and DBpedia include 4 and 14 classes, respectively, for topic classification. Amazon and Yelp both consist of 5 classes for sentiment classification, while Yahoo Answers contains 10 classes for question-and-answer classification.

For our experiments, we closely follow Wang et al. (2024). We use a train-test split of $115\,000$ and $7\,600$ samples, respectively, for all experiments involving BERT. We train the model using five different task orders. For the T5 and Llama experiments, in line with prior research, we use 4 of the 5 tasks, excluding Yelp, with 16 samples for training while keeping the test set unchanged. These experiments are conducted with three different task orders. Following previous work, we report macro-F1 score on the AfriSenti dataset and macro-accuracy on WOS and MTL5 datasets (Wang et al., 2024; Muhammad et al., 2023). We omit evaluations of backward transfer as all evaluated methods fully prevent catastrophic forgetting through architectural design. Forward transfer is evaluated by comparing the final performance to a model only trained on one task, thus showcasing knowledge sharing potential. Used task orders for all datasets are listed in Appendix A.3.

To evaluate whether the effectiveness of our method extends to a larger number of tasks, we conduct an additional experiment on longer sequences. Following Razdaibiedina et al. (2023), we use the MTL15 dataset, which consists of 15 classification tasks.

## 4.2 BACKBONE LMS AND PEFT TYPE

As our experiments follow Wang et al. (2024), we also use the encoder-based BERT-base (Devlin et al., 2019) and AfroXLMR (Alabi et al., 2022) as pre-trained backbones for the WOS and AfriSenti datasets, respectively. For experiments with the MTL5 dataset, we employ the encoder-based BERT-base, the encoder-decoder-based T5 (Raffel et al., 2020), and the decoder-based LLaMA2-7B (non-instruct) models (GenAI, 2023). In our experiments with BERT, AfroXLMR, and T5, we utilize prefix tuning, while for LLaMA2, we apply LoRA.

## 4.3 BASELINES

**Sequential Full-FT**: The model parameters are fully trainable and the entire model is trained on each task sequentially.

**PerTask-PEFT**: The backbone model parameters are frozen and a dedicated PEFT module is trained for each task separately.

**Sequential PEFT-FT**: The backbone parameters are frozen and a single PEFT module is trained sequentially on all tasks.

**EPI** (Wang et al., 2023): Trains a dedicated PEFT module and task representation vector for each task. There is no knowledge transfer between previously learned tasks and the current task.

**ProgPrompt** (Razdaibiedina et al., 2023): A parameter isolation-based method that conducts task-specific prefix tuning. To facilitate knowledge transfer, prefix modules from previous tasks are concatenated with the prefix of the current task.

**MoCL** (Wang et al., 2024): Trains a dedicated PEFT module and task feature representation vector for each task. To facilitate knowledge transfer, similarity scores between the current task and previously learned tasks are calculated based on the task feature vectors. The PEFT module for the current task is then initialized as a weighted sum of the PEFT modules from similar previous tasks.

### 4.4 IMPLEMENTATION DETAILS

For our experiments, we use the same maximum sequence length, prefix length, and LoRA rank as Wang et al. (2024). We employ the AdamW optimizer (Loshchilov, 2017) and a batch size of 8. In experiments with expandable PEFTs, the initial prefix length is set to 8 for AfriSenti, 10 for MTL15, 16 for WOS, 20, and 50 for MTL5 respectively. The LoRA rank is set to 4. Unless otherwise specified, the PEFT module is expanded by 25% of its initial size during one expansion step. We initialize LoRA's $A$ matrix and $A_{exp}$ using Kaiming uniform distribution (He et al., 2015) and LoRA's $B$ and $B_{exp}$ with zeros following Hu et al. (2021). The prefix $P$ and $P_{exp}$, are initialized using $\mathcal{U}(0, 1)$. The task embeddings $e^{(t)}$ and $e_{exp}^{(t)}$, are initialized using $\mathcal{N}(0, 1)$. For LoRA we extend the intermediate rank and for prefix-tuning we add entire prefix tokens. We use early stopping with patience of 5 steps. Detailed hyperparameter information for each experiment can be found in Appendix A.4.

## 5 RESULTS

**Full Continual Learning** In this experiment, we compare GE-PEFT integrated into AfroXLMR and BERT on two near-domain datasets, AfriSenti and WOS, as well as the far-domain dataset MTL5, against various baselines and state-of-the-art CL approaches, as summarized in Table 1. In the near-domain experiments, non-CL solutions like sequential full finetuning (Seq Full-FT) show a strong negative impact, which can be attributed to catastrophic forgetting. Using separate PEFT modules for each task (PerTask-PEFT) mitigates catastrophic forgetting and outperforms sequential PEFT finetuning (Seq PEFT-FT) across all datasets, as well as ProgPrompt and EPI on AfriSenti. Notably, only MoCL and our GE-PEFT show improvements over this baseline, leveraging knowledge sharing across closely related tasks. Overall, across all near-domain datasets, our proposed architecture, GE-PEFT, outperforms MoCL, the current state-of-the-art, by a large margin. In the far-domain experiment on MTL5, we observe similar behavior, though the differences compared to the per-task baseline (PerTask-PEFT) are smaller than those observed on WOS. This is expected, as less knowledge from previous tasks can be leveraged through knowledge sharing. Notably, GE-PEFT still outperforms all baselines and state-of-the-art approaches, likely due to its ability to gate irrelevant information as later analyzed in the ablation study.

Table 1: Near domain results on AfriSenti and WOS data with similar tasks and far domain results on MTL5 with prefix tuning, using AfroXMLR on AfriSenti and BERT-base on WOS and MTL5. Results averaged over 3 seeds. [†] indicates results taken from Wang et al. (2024).

| Dataset | AfriSenti - F1 | | | | WOS - Acc. | MTL5 - Acc. | | | | |
| Model | Avg | Seq1 | Seq2 | Seq3 | Avg | Avg | Seq1 | Seq2 | Seq3 | Seq 4 |
|---|---|---|---|---|---|---|---|---|---|---|
| Seq Full-FT[†] | 6.2 | 5.6 | 6.5 | 6.3 | 47.2 | 14.8 | 27.8 | 26.7 | 4.5 | 18.4 |
| Seq PEFT-FT[†] | 49.1 | 50.1 | 49.7 | 47.5 | 53.9 | 66.5 | 66.4 | 65.7 | 65.4 | 68.5 |
| PerTask-PEFT[†] | 52.4 | 52.4 | 52.4 | 52.4 | 82.8 | 77.6 | 77.6 | 77.6 | 77.6 | 77.6 |
| ProgPrompt[†] | 49.1 | 50.2 | 46.7 | 50.3 | 89.9 | 77.9 | 78.0 | 77.9 | 77.9 | 77.9 |
| EPI[†] | 43.1 | 41.5 | 42.7 | 45.2 | 77.8 | 77.3 | 77.4 | 77.3 | 77.2 | 77.4 |
| MoCL | 58.0 | 58.5 | 56.3 | 59.0 | 90.3 | 78.4 | 78.6 | 78.5 | 78.0 | 78.3 |
| GE-PEFT (ours) | **62.1** | **62.7** | **62.3** | **61.3** | **91.5** | **79.4** | **79.5** | **79.4** | **79.4** | **79.3** |

Table 2: Far domain results on dissimilar tasks in a few-shot setting with 16 training samples on the MTL5 dataset, averaged over 3 seeds.

(a) Accuracy for prefix tuning on T5.

| Model | Avg | Seq1 | Seq2 | Seq3 |
|---|---|---|---|---|
| Seq Full-FT[†] | 28.5 | 18.9 | 24.9 | 41.7 |
| Seq PEFT-FT[†] | 27.2 | 24.6 | 30.3 | 25.0 |
| PerTask-PEFT[†] | 75.1 | 75.1 | 75.1 | 75.1 |
| ProgPrompt[†] | 75.1 | 75.0 | 75.0 | 75.1 |
| EPI[†] | 56.4 | 49.7 | 54.1 | 65.3 |
| MoCL[†] | 75.9 | 75.6 | 75.4 | 76.7 |
| GE-PEFT (ours) | **77.3** | **77.3** | **77.3** | **77.2** |

(b) Accuracy for LoRA on Llama 2.

| Model | Avg | Seq1 | Seq2 | Seq3 |
|---|---|---|---|---|
| Seq PEFT-FT | 23.5 | 24.7 | 20.8 | 25.0 |
| PerTask-PEFT | 73.2 | 73.2 | 73.2 | 73.2 |
| EPI[†] | 48.4 | 48.1 | 48.0 | 49.0 |
| MoCL | 73.8 | 74.1 | 73.6 | **73.8** |
| GE-PEFT (ours) | **74.2** | **74.8** | **74.4** | 73.3 |

**Few-Shot Continual Learning**   The results in the few-shot CL setting, are shown in Table 2a for the prefix-tuning based T5 and in Table 2b for LoRA based Llama 2. Our results with T5 align with our previous findings. While sequential full finetuning (Seq Full-FT) and sequential PEFT finetuning (Seq PEFT-FT) exhibit the lowest performance, GE-PEFT consistently outperforms all baselines and state-of-the-art models. Among these, MoCL and ProgPrompt perform similarly to PerTask-PEFT, whereas EPI shows mediocre results. Applying GE-PEFT to LoRA PEFT modules of Llama 2 outperforms all baselines and EPI. We observe a slight performance boost compared to MoCL for two of the three task sequences, while being more parameter-efficient, since we do not incrementally introduce task-specific PEFT modules. Particularly, for this parameter setting and dataset, we require only $\approx 34\%$ of the trainable parameters of MoCL. To further highlight the performance differences between GE-PEFT and MoCL, we include additional sequences in Appendix A.1. Overall, our results indicate that knowledge sharing still provides benefits in few-shot settings, even on far domain data where knowledge does not easily transfer across tasks.

**Long Sequence Few-Shot Continual Learning**   We additionally evaluate performance on long sequences with 15 tasks on the MTL15 dataset. Results in Table 3 show that GE-PEFT outperforms all other approaches on all sequences while maintaining a considerably higher parameter efficiency with fewer total prefixes used. Growing of GE-PEFT also provides small improvements over the gated non-growing adapter variant G-PEFT, while still maintaining a higher parameter efficiency then MoCL and ProgPrompt that must maintain individual prefixes for all previous tasks.

**Ablation Study**   As we introduce two components in our GE-PEFT approach, the gating (G) and the parameter expansion (E), we analyze the benefit of each individual component in this ablation study. As G-PEFT uses gating to train multiple tasks into the same PEFT module, it avoids catastrophic forgetting, however is prone to parameter saturation. E-PEFT, on the other hand, can prevent saturation by the expansion mechanism, although no mechanics actively prevent catastrophic forgetting in multi-task training. We also report the current state-of-the-art, MoCL for comparison. The results are shown in Table 4 for full CL with prefix tuning on BERT and AfroXLMR. G-PEFT performs mostly slightly behind the GE-PEFT strategy with the exception of one sequence in AfriSenti and two sequences in MTL5 but consistently surpasses the MoCL method across all sequences and datasets. Without gating, E-PEFT performs noticeably worse across multiple sequences, highlight-

Table 3: Accuracy and final prefix sizes for prefix tuning of T5 on MTL5 with 15 tasks using 20 training samples, macro-averaged over 3 seeds. Comparison with using only the gating (G) component component of GE-PEFT, MoCL and ProgPrompt as SOTA.

| Model | Avg | Seq1 | Seq2 | Seq3 | Prefixes per Task | Prefixes Total |
|---|---|---|---|---|---|---|
| MoCL | 70.2 | 69.6 | 70.2 | 70.9 | 10 | 150 |
| ProgPrompt | 72.4 | 72.2 | 73.0 | 72.2 | 10 | 150 |
| G-PEFT | 73.0 | 73.1 | 72.8 | 73.2 | 10 | 10 |
| GE-PEFT | **73.3** | **73.2** | **73.1** | **73.7** | 27 | 27 |

Table 4: BERT ablation study with prefix tuning. Results averaged over 3 seeds. Comparison with using only the gating (G) component or the expansion (E) component of GE-PEFT. MoCL as SOTA for comparison.

| Dataset | AfriSenti - F1 | | | | WOS - Acc. | MTL5 - Acc. | | | | |
|---------|------|------|------|------|------------|------|------|------|------|------|
| Model | Avg | Seq1 | Seq2 | Seq3 | Avg | Avg | Seq1 | Seq2 | Seq3 | Seq4 |
| MoCL | 58.0 | 58.5 | 56.3 | 59.0 | 90.3 | 78.4 | 78.6 | 78.5 | 78.0 | 78.3 |
| E-PEFT | 58.6 | 56.2 | 59.3 | 60.2 | 75.4 | 65.6 | 62.4 | 65.6 | 65.6 | 68.8 |
| G-PEFT | 61.1 | 60.8 | 60.9 | **61.5** | 91.3 | **79.4** | 79.4 | 79.3 | **79.6** | **79.4** |
| GE-PEFT | **62.1** | **62.7** | **62.3** | 61.3 | **91.5** | **79.4** | **79.5** | **79.4** | 79.4 | 79.3 |

ing the risk of catastrophic forgetting when all tasks are repeatedly trained into a single PEFT module and the importance of our gating component.

Our ablation on T5 with prefix tuning, summarized in Table 5a, reveals that both, G-PEFT and GE-PEFT consistently outperform MoCL. However, the expansion component (GE) does not provide additional benefits over our gated shared PEFT approach (G). While GE-PEFT falls slightly behind G-PEFT on one sequence, both models perform on a par in the others. E-PEFT performs even worse compared to our previous experiments.

Table 5b shows, that for Llama, the impact of catastrophic forgetting for E-PEFT is even larger, as indicated by the results falling even below the sequential PEFT finetuning baseline (cf. Table 2b). GE-PEFT, G-PEFT, and MoCL each achieve the highest accuracy on one task sequence, indicating that there is no clear winner in this comparison. Notably, while GE-PEFT, G-PEFT, and MoCL deliver comparable performance, G-PEFT and GE-PEFT are more parameter-efficient. G-PEFT requires a total of 3 146 752 trainable parameters (with a small additional fraction for expansion in GE-PEFT, cf. Table 7), whereas MoCL requires 2 363 648 parameters for each of the four tasks.

**Parameter Analysis** Our method controls parameter efficiency in the growing PEFT module through two mechanisms: $\lambda$, which promotes sparse neuron usage during task training, and the expansion size, a hyperparameter that controls the network's growth rate. We vary the regularization value $\lambda$, with high $\lambda$ values indicating a high penalty for reserving many neurons. The expansion size, which determines how many additional neurons are added to the PEFT module during the growth step, is defined as a percentage of the initialization size of the module. All runs are conducted using the first sequence order (Seq1) of the respective dataset. For all models, we report the used-parameter quota, indicating the percentage of neurons reserved after training all tasks. For prefix-based models, the final prefix size is shown in brackets (cf. Table 6). For LoRA, the expansion size (cf. Table 7) indicates the total number of additional neurons added to the PEFT modules.

Table 6 shows the performance of BERT on AfriSenti as a near-domain dataset with very similar tasks that can benefit from each other. As a result, the model does not require all parameters to achieve good performances and only sparsely expands when necessary. Still, very high regularization through $\lambda$ results in poor performance as the model restricts itself to very few parameters, independent of expansion sizes. Lower regularization values allows the model to reserve more parameters and improve performance. As the model blocks only a small number of neurons needed

Table 5: Far domain ablation study on dissimilar tasks in a few-shot setting with 16 training samples on MTL5, averaged over 3 seeds. Comparison with using only the gating (G) component or the expansion (E) component of GE-PEFT. MoCL as SOTA for comparison.

(a) Accuracy for prefix tuning on T5.

| Model | Avg | Seq1 | Seq2 | Seq3 |
|-------|------|------|------|------|
| MoCL | 75.9 | 75.6 | 75.4 | 76.7 |
| E-PEFT | 26.5 | 24.6 | 29.6 | 25.2 |
| G-PEFT | **77.4** | **77.6** | 77.3 | 77.2 |
| GE-PEFT | 77.3 | 77.3 | **77.3** | **77.2** |

(b) Accuracy for LoRA on Llama 2.

| Model | Avg | Seq1 | Seq2 | Seq3 |
|-------|------|------|------|------|
| MoCL | 73.8 | 74.1 | 73.6 | **73.8** |
| E-PEFT | 22.8 | 24.2 | 21.0 | 23.2 |
| G-PEFT | **74.4** | 74.6 | **74.8** | 73.7 |
| GE-PEFT | 74.2 | **74.8** | 74.4 | 73.3 |

Table 6: Parameter Analysis for BERT with prefix tuning on AfriSenti.

| Avg. | F1 | | | | Used-Parameter Quota (Prefix Size) | | | |
|---|---|---|---|---|---|---|---|---|
| $\lambda$\Expansion | 25% | 50% | 75% | 100% | 25% | 50% | 75% | 100% |
| 100 | 56.66 | 57.03 | 56.30 | 58.19 | 0.21 (13) | 0.19 (12) | 0.19 (12) | 0.08 (19) |
| 10 | 60.32 | 60.87 | 59.21 | 60.93 | 2.19 (15) | 3.32 (20) | 1.90 (20) | 1.49 (11) |
| 1 | 62.71 | 60.32 | 59.62 | 60.71 | 17.42 (13) | 18.84 (20) | 14.76 (24) | 14.76 (11) |
| 0 | 60.92 | 61.20 | 60.71 | 62.00 | 50.03 (12) | 49.97 (20) | 49.97 (24) | 49.93 (13) |
| 0.1 | 62.41 | 62.40 | 62.05 | 62.08 | 22.89 (14) | 23.03 (19) | 20.93 (18) | 22.19 (16) |

for similar tasks, the expansion size has minimal impact on performance. This demonstrates the information transfer capabilities of GE-PEFT on highly related tasks.

Next we evaluate parameters in the far-domain setting, where tasks do not necessarily benefit from each other. Here, the Llama 2 model on the low-resource MTL5 dataset in Table 7 requires a considerably larger amount of parameters, reserving large amounts of neurons even on higher $\lambda$ values. Both the regularization and the expansion parameters influence the number of used parameters. Increasing regularization reduces the used-parameter quota as $\lambda$ increases, while a larger expansion size leads to a noticeable increase in LoRA size by the end of training.

Further experiments for BERT with prefix tuning on the WOS dataset and T5 with prefix tuning on the MTL5 dataset can be found in Appendix A.2. Experiments in these configurations suggest that the models generally have sufficient parameters to encode the information, though a balanced regularization parameter $\lambda$ is required to prevent arbitrary growth when unused neurons are still available.

Overall, our results indicate that model parameters can be influenced by regularization and expansion size, while the sensitivity to a specific hyperparameter appears to be model and data dependent. Results also show that accuracy is not directly dependent on either parameter, indicating that small amounts of expansion size and a medium size of regularization still result in high performance, which further aids GE-PEFT's parameter-efficiency.

# 6 CONCLUSION

In this work, we introduced Gated Expandable Parameter-Efficient Fine-Tuning (GE-PEFT), a novel approach for CL in LMs that effectively addresses four key CL criteria: catastrophic forgetting prevention, parameter efficiency, knowledge transfer, and managing model capacity. By integrating a single, dynamically expanding PEFT module with a gating mechanism, GE-PEFT enables continuous knowledge transfer throughout training and inference while maintaining task separation. Our experimental results across multiple task-incremental CL benchmarks demonstrate that GE-PEFT consistently outperforms existing state-of-the-art methods. Yet, although our GE-PEFT approach is effective for task-incremental learning, it has not yet been evaluated for class-incremental learning scenarios where the task of a given input is not known at inference time. Further, our current version of GE-PEFT relies on simple heuristics for its expansion strategy. Here, more complex and effective expansion strategies are a promising area of future work.

Table 7: Parameter Analysis for Llama with LoRA on MTL5.

| Avg. | Accuracy | | | | Used-Parameter Quota (LoRA Expansion Size) | | | |
|---|---|---|---|---|---|---|---|---|
| $\lambda$\Expansion | 25% | 50% | 75% | 100% | 25% | 50% | 75% | 100% |
| 1000 | 73.22 | 74.83 | 74.45 | 74.54 | 91.21 (128) | 90.67 (213) | 91.46 (192) | 91.20 (512) |
| 100 | 73.32 | 73.07 | 74.50 | 73.84 | 91.50 (192) | 91.59 (85) | 91.26 (320) | 91.70 (341) |
| 10 | 74.79 | 74.34 | 73.93 | 74.98 | 92.34 (21) | 92.28 (171) | 92.32 (256) | 91.93 (341) |
| 1 | 73.40 | 73.86 | 74.04 | 73.73 | 92.83 (85) | 92.64 (192) | 92.91 (192) | 93.03 (384) |
| 0 | 73.74 | 73.90 | 74.41 | 73.37 | 93.77 (85) | 93.77 (85) | 93.75 (384) | 93.75 (341) |

REPRODUCIBILITY STATEMENT

All datasets used throughout this study are publicly available. To facilitate direct comparisons to existing and future works, we use the fixed data splits established in literature. The evaluated sequences of tasks are taken directly from previous work for comparability (Wang et al., 2024) and are listed again in the appendix for completeness. Further, the hyperparameters used within this study such as PEFT and optimizer parameters are listed in the appendix. Lastly, to facilitate open research and reproducibility, we provide the code and data of our experiments for review and make it publicly available `https://shorturl.at/lxidM`.

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

# A APPENDIX

## A.1 ADDITIONAL TASK SEQUENCES FOR FAR DOMAIN EXPERIMENTS

To better highlight the differences between GE-PEFT and the closest baseline, MoCL, on MTL5 with Llama 2, we provide results for additional task sequences. Results in Table 8 show that GE-PEFT consistently outperforms MoCL in all but one sequence.

Table 8: Accuracy for additional task sequences for the far domain experiments in Table 2b, using 16 training samples on MTL5 with LoRA on Llama 2 and averaging over 3 seeds. MoCL as SOTA for comparison.

| Model | Total Avg | Seq1 | Seq2 | Seq3 | Seq4 | Seq5 | Seq6 | Seq7 |
|---|---|---|---|---|---|---|---|---|
| MoCL | 73.3 | 74.1 | 73.6 | **73.8** | 72.6 | 73.2 | 72.4 | 73.5 |
| GE-PEFT | **74.1** | **74.8** | **74.4** | 73.3 | **74.7** | **73.4** | **74.0** | **74.1** |

## A.2 ADDITIONAL PARAMETER ANALYSIS EXPERIMENTS

Here we provide further parameter analysis experiments to complement the presented results in Section 5.

For the BERT-base model in Table 9, results show slight fluctuations of accuracy independent of the choices of $\lambda$ and expansion. The model also appears to be insensitive to regularization as increasing the $\lambda$ parameter shows only a slight trend of decreasing parameter usage with multiple outliers with high $\lambda$, e.g., with 25% expansion rate. Despite high $\lambda$ values, the model maintains a high used-parameter quota, suggesting that the small BERT-base model needs parameters for task-solving, limiting the effect of regularization. Increases in expansion size on the other hand result in more taken parameters, indicated by increased prefix size. However, as high accuracy is also achievable with low expansion sizes, this indicates that fast growth is not necessary to reach high performance.

In contrast to the BERT-base results, the larger T5 model on the low-resource MTL5 dataset in Table 10 shows a more pronounced effect of regularization that reduces the used-parameter quota with higher $\lambda$ values. Expansion rate does, however, only show small impacts on this model and dataset combination, as only few expansion steps are made by the model in all configurations. Especially in combination with the regularization, this suggests a behavior where the available parameters are sufficient to encode all information. Low regularization appears to result in faster saturation of all available neurons which does not require growing steps, while high regularization with small expansion size causes multiple growing steps as the regularization prevents the direct use of the available neurons.

Table 9: Parameter Analysis for BERT with prefix tuning on WOS.

| Avg. | Accuracy | | | | Used-Parameter Quota (Prefix Size) | | | |
|---|---|---|---|---|---|---|---|---|
| $\lambda$\Expansion | 25% | 50% | 75% | 100% | 25% | 50% | 75% | 100% |
| 100 | 91.20 | 91.22 | 91.37 | 91.50 | 94.45 (21) | 84.01 (53) | 90.68 (64) | 86.40 (80) |
| 10 | 91.54 | 91.31 | 91.45 | 91.14 | 91.40 (26) | 96.53 (29) | 94.79 (32) | 93.55 (59) |
| 1 | 91.62 | 91.22 | 91.50 | 91.53 | 91.56 (25) | 88.63 (48) | 87.40 (44) | 93.82 (69) |
| 0.1 | 90.87 | 91.35 | 91.18 | 91.33 | 96.79 (20) | 87.89 (48) | 87.60 (64) | 93.77 (59) |
| 0.01 | 91.17 | 91.21 | 91.27 | 91.03 | 96.46 (24) | 85.90 (32) | 93.00 (52) | 90.94 (59) |
| 0 | 91.56 | 91.36 | 91.30 | 91.40 | 96.72 (27) | 87.29 (37) | 92.70 (52) | 91.33 (48) |

Table 10: Parameter Analysis for T5 with prefix tuning on MTL5.

| Avg. | Accuracy | | | | Used-Parameter Quota (LoRA Expansion Size) | | | |
|---|---|---|---|---|---|---|---|---|
| $\lambda$\Expansion | 25% | 50% | 75% | 100% | 25% | 50% | 75% | 100% |
| 100 | 76.52 | 75.78 | 76.70 | 75.86 | 16.29 (58) | 17.01 (50) | 8.90 (50) | 19.77 (50) |
| 10 | 77.34 | 77.57 | 77.32 | 77.06 | 42.37 (66) | 47.91 (75) | 45.38 (50) | 45.89 (50) |
| 1 | 77.59 | 77.69 | 77.44 | 76.87 | 71.83 (90) | 66.98 (83) | 74.67 (50) | 76.16 (50) |
| 0 | 77.12 | 77.07 | 77.43 | 76.84 | 82.69 (74) | 76.03 (67) | 77.74 (62) | 88.54 (67) |

## A.3 DATASET DETAILS

The task orders used for all datasets in our experiments are detailed in Table 11 for reproducibility and chosen after Wang et al. (2024) where applicable.

Table 11: The different orders of task sequences used for incremental task learning experiments following Wang et al. (2024).

| Dataset | Model | Task Sequence |
|---|---|---|
| | BERT | ag → yelp → amazon → yahoo → db |
| | BERT | yelp → yahoo → amazon → db → agnews |
| | BERT | db → yahoo → ag → amazon → yelp |
| MTL5 | BERT | yelp → ag → db → amazon → yahoo |
| | T5, Llama 2 | db → amazon → yahoo → ag |
| | T5, Llama 2 | db → amazon → ag → yahoo |
| | T5, Llama 2 | yahoo → amazon → ag → db |
| | Llama 2 | ag → yahoo → amazon → db |
| | Llama 2 | amazon → ag → yahoo → db |
| | Llama 2 | ag → db → yahoo → amazon |
| | Llama 2 | amazon → yahoo → db → ag |
| MTL15 | T5 | mnli → cb → wic → copa → qqp → boolqa → rte → imdb → yelp → → amazon → sst2 → db → ag →multirc → yahoo |
| | T5 | multirc → boolqa → wic → mnli → cb → copa→ qqp → rte → imdb → sst2 → db → ag → yelp → amazon → yahoo |
| | T5 | yelp → amazon → mnli → cb → copa → qqp → rte → imdb → sst2 → db → ag → yahoo → multirc → boolqa → wic |
| | AfroXLMR | am → dz → ha → ig → kr → ma → pcm → pt → sw → ts → twi → yo |
| AfriSenti | AfroXLMR | ma → pcm → kr → pt → ig → sw → ha → ts → dz → twi → am → yo |
| | AfroXLMR | am → dz → ha → ma → ig → kr → sw → ts → twi → yo → pcm → pt |
| WOS | BERT | 1 → 2 → 3 → 4 → 5 → 6 → 7 |

## A.4 DETAILED HYPERPARAMETERS

Detailed hyperparameters for all experiments can be found in Table 12.

Table 12: Hyperparameter settings for WOS-BERT, AfriSenti-AfroXLMR, and MTL5-BERT models following Wang et al. (2024).

| Hyperparameters | WOS BERT | AfriSenti AfroXLMR | MTL5 BERT | MTL5 T5 | MTL5 Llama2 |
|---|---|---|---|---|---|
| Epochs | 40 | 40 | 40 | 40 | 40 |
| Early stop patience | 5 | 5 | 5 | 5 | 5 |
| Learning rate | 1e-5 | 2e-4 | 1e-5 | 2e-2 (yahoo, db) 5e-2 (others) | 1e-3 |
| Max. sequence len. | 256 | 128 | 256 | 512 | 512 |
| Prefix len., rank | 16 | 8 | 20 | 50 | 4 |
| Batch size | 8 | 8 | 8 | 8 | 8 |