# OpenReview forum: "GE-PEFT: Gated Expandable Parameter-Efficient Fine-Tuning for Continual Learning"
_ICLR.cc/2025/Conference — Submitted to ICLR 2025_

### Official Review · Reviewer_FUVm · 2024-11-02

**Soundness:** 2
**Presentation:** 3
**Contribution:** 2
**Rating:** 5
**Confidence:** 4

**Summary:**

This paper proposes Gated Expandable Parameter-Efficient Fine-Tuning (GE-PEFT), a framework designed for continual learning in LLMs to address catastrophic forgetting. GE-PEFT utilizes a single, expandable PEFT module with a gating mechanism to memorize task-specific knowledge and dynamically adjusts its size for new tasks. Experiment results across multiple benchmarks show that GE-PEFT shows good performance.

**Strengths:**

1. The proposed approach is well-motivated.
2. The proposed approach is novel.
3. The proposed approach show good performance.

**Weaknesses:**

1. It would be better to compare model performance on the benchmark datasets before and after training, so the readers will learn the effect of the learning process on previously learnt task. Currently, the authors just report the performance after training. It would be a plus if the authors can include the model performance before training.
2. Although the overall approach is novel, some techniques, such as gate, LoRA, prefix tuning are not new.
3. It would be better to have a case study.

**Questions:**

1. Which version of LLaMA-2 does the paper use? The base version or the instruct version?

---

> ### Author Response · Authors · 2024-11-28
>
> We thank the reviewer for their insightful feedback and wish to address it as follows:
> - **Model performance before training.** We have initially tested the generative models T5 and LLaMA-2 without fine-tuning on the datasets but noted that the models were not able to identify any of the solutions correctly, giving accuracies of 0%.
> - **Some techniques are not new.**
> We build on prior works in continual learning, parameter-efficient fine-tuning, and growing networks, but emphasize that the unique combination of these approaches in GE-PEFT has not been explored before. Our method not only addresses key challenges in continual fine-tuning but also exceeds the current state-of-the-art in this application area.
> - **Case study.** We have expanded both our ablation study and parameter efficiency evaluation with additional experiments according to suggestions from the other reviewers, showcasing the capabilities of GE-PEFT on further datasets.
> - **LLaMA-2 version.** We use the LLaMA-2 base version following previous work. We added this clarification to the paper.

---

### Official Review · Reviewer_AuY5 · 2024-11-04

**Soundness:** 3
**Presentation:** 3
**Contribution:** 2
**Rating:** 5
**Confidence:** 3

**Summary:**

This paper proposes a method called GE-PEFT for solving continuous learning problems. This method addresses four key CL criteria simultaneously by integrating a dynamically expanding PEFT module and a gating mechanism: preventing forgetting, parameter efficiency, knowledge transfer, and managing model capacity. The experimental results show that GE-PEFT performs well in multiple-task incremental CL benchmark tests and is more effective than the best existing methods. In addition, the author provides code and data to promote open research and reproducibility.

**Strengths:**

A reasonable continuous learning approach is proposed, which employs dynamically expanding shared parameters and a gating mechanism to solve the problems of parameter efficiency and knowledge migration in continuous learning.

**Weaknesses:**

1.Although the model design is reasonable, it seems an ensemble of existing methods such as gate mechanism and mask technique.

2.The difference in performance between G-PEFT and GE-PEFT in the ablation experiment is insignificant, and the effectiveness of the E module has not been verified in the experiment.

3.This paper needs a detailed theoretical analysis of the proposed GE-PEFT methodology. For example, it is questionable whether expandable LoRA can effectively update the parameters in LLMs.

4.Most comparison methods are PEFT and its variants, it would be better to include more relevant baselines. Besides, it would be better to evaluate each method with the 4 criteria that the proposed method has well addressed as claimed.

**Questions:**

The paper is generally well-written, while the paper needs more explanation about the difference and novel improvements between PEFT and GE-PEFT.

---

> ### Author Response · Authors · 2024-11-28
>
> We thank reviewer AuY5 for their valuable feedback. We have restructured and extended both the ablation study and the parameter efficiency study with further experiments to better highlight the unique benefits of GE-PEFT. Our results show that balancing expansion and regularization is beneficial in complex and dissimilar tasks that do not allow for much knowledge sharing. On more simple tasks where language model parameters are easily sufficient, the E module of GE-PEFT is less effective. Nevertheless, we believe that this dynamically adaptable architecture provides significant benefits in continual applications.
>
> Regarding our evaluation, we wish to clarify that we explore all 4 criteria that GE-PEFT addresses: Prevention of catastrophic forgetting is guaranteed in all methods due to their architecture and observable when accuracies fall below those of independent fine-tuning on separate tasks. Parameter efficiency is evaluated through our parameter efficiency study, which we have extended to accommodate an additional dataset. Knowledge transfer is observable through accuracies above the values achieved when using direct fine-tuning on individual tasks. Lastly, expansion is achieved by design and shown in our ablation study, while baselines in the area of continual fine-tuning are not capable of expanding at all.

---

### Official Review · Reviewer_L1dz · 2024-11-04

**Soundness:** 3
**Presentation:** 2
**Contribution:** 2
**Rating:** 5
**Confidence:** 3

**Summary:**

The paper introduces the Gated Expandable PEFT module, which improves continual learning by effectively addressing knowledge transfer, preventing catastrophic forgetting, and enhancing parameter efficiency and scalability.  This method can be applied to both LoRA and prefix tuning. The authors implement a dynamic adjustment mechanism for model size to align with the needs of current tasks while retaining previously acquired knowledge. Experimental results across various models demonstrate that the proposed method outperforms several baseline methods.

**Strengths:**

1. The authors effectively utilize the Gated PEFT module in conjunction with an expandable parameters strategy, clearly motivated to address the catastrophic forgetting in continual learning for PEFT.

2. The proposed method can be implemented in both LoRA and prefix tuning.

3. The proposed method outperforms several baseline methods across four different language models.

**Weaknesses:**

1. The Gated layer method employed has been utilized in previous CL work, while the additional expandable parameters strategy has little effect (as shown in Table 3 and 4).

2. Tables 5 to 7 are difficult to read and should be presented in figure.

Minor:

3. Using subsections to structure Section 4 would improve its readability.

4. Reporting the standard deviation of results averaged over random seeds would enhance clarity.

**Questions:**

Why does the expandable parameters strategy provide little additional benefit in many cases? Given its complexity and the extra parameters required, its inclusion seems to lack sufficient justification.

I would appreciate it if the authors could address the concerns or clarify if I have misunderstood any part of the paper.

---

> ### Author Response · Authors · 2024-11-28
>
> We thank reviewer L1dz for their valuable feedback and subsequently decided to restructure both the ablation study and the parameter efficiency study to better highlight the benefits of GE-PEFT. We additionally provide further experiments in these sections as suggested by Reviewer x7yT. Our results show that while some combinations of dataset and language model do not require much expanding due to the already sufficient number of parameters for the given task complexity, on more complex tasks a balance of expansion size and regularization is crucial to achieving high performance and parameter efficiency.

---

### Official Review · Reviewer_x7yT · 2024-11-06

**Soundness:** 2
**Presentation:** 2
**Contribution:** 2
**Rating:** 3
**Confidence:** 5

**Summary:**

In this paper the authors propose a method for parameter-efficient finetuning, GE-PEFT that is specifically designed for the (task-incremental) continual learning setting. It works by combining two components: a gating mechanism that consists of a learned, per-task binary mask over the PEFT weights (to minimize catastrophic forgetting while allowing for parameter sharing in the PEFT weights), and expansion of the PEFT weights (in order to provide sufficient capacity as new tasks are added). They demonstrate the approach on two PEFT methods: LoRA and prefix tuning, on text classification benchmarks: AfriSenti (sentiment analysis on 12 African languages), Web-of-Science (document classification, near domain) and MTL5 (document classification, far domain) benchmarks. They experiment with BERT, T5 and Llama models, in full and few-shot settings, although the full cross product of these experiments is not considered. They show that their proposed method out-performs other CL PEFT methods in terms of final overall micro-averaged accuracy after sequential training.

**Strengths:**

- **Interesting problem setting.** Continual learning, and specifically parameter-efficient continual learning, is an important and realistic problem setting.
- **Paper is well-written.** The paper was reasonably well-written and easy to understand.
- **Simple method.** The method is relatively straightforward to implement and does not add too many new hyperparameters; the most finicky part seems to be the process for determining expansion, but the expansion doesn’t seem to be needed in many cases anyway.

**Weaknesses:**

- **Insufficient comparison to baseline CL methods.** In experimental results, the authors compare to recent state-of-the-art methods specific to parameter-efficient CL, but not state-of-the-art CL methods more broadly, or even old but effective methods such as EWC and episodic replay. As far as I can tell there is no reason that those methods would not also work for PEFT. In the related work section in the appendix, justification is given based on the fact that replay requires keeping examples around, and that both strategies do not *entirely* eliminate catastrophic forgetting. But it would make sense to actually demonstrate that this is the case, especially since the experimental setting in the paper does not involve many tasks (which would mitigate the issue of keeping examples around for each task, for example). However, forgetting is not even evaluated in experimental results. Optimization-based approaches such as MC-SGD [(Mirzadeh et al. 2021)](https://arxiv.org/abs/2010.04495) are not even mentioned.
- **Unclear and insufficient evaluation metrics.** It guess the numbers reported in the tables are average accuracy across all tasks in the test set after sequential training? It’s not clear whether the authors are micro- or macro-averaging across tasks. It’s not clearly stated, and there are many ways to evaluate aspects of performance in the task-incremental setting beyond average accuracy: you might also measure forgetting (backward transfer), and task accuracy (forward transfer). This way one can get a deeper sense for how different approaches trade off these aspects of CL performance, and better support the claims in the paper. See [Lopez-Alt & Ranzato (2017)](https://proceedings.neurips.cc/paper/2017/hash/f87522788a2be2d171666752f97ddebb-Abstract.html) for further discussion/definitions.
- **Insufficient analysis of results.** There are not compelling experiments or other analysis explaining the experimental results. For example, adding the expansion component only seems to help in a single experimental setting (BERT+AfriSenti), and does not seem to provide much in the other settings. Why is this the case? When should a user bother with that aspect of the model?
- **Experimental setup follows previous work, but is simply not that compelling.** The task-incremental CL setting with a small number of tasks (5-12 in this case, with 12 actually being the same task, but different languages) is a standard CL benchmark, but it’s just not that realistic of a CL setting (versus natural distribution shifts, such as temporal or domain shift.) This limits the real-world applicability of the approach. This work does not experiment with longer diverse task sequences, such as the 15 task sequence benchmark used in baselines from Wang et al. and Razdaibiedina et al. Additionally, very few task orderings are experimented with, even though for some experiments there are only 4-5 tasks and therefore 4! = 24 or 5! = 120 possible task orderings. In those cases, especially since the proposed method is PEFT, evaluating every task ordering seems feasible, or at least more than 3, which seems like it would be needed to obtain statistically significant results. This would improve robustness of experimental results.
- **No/limited efficiency evaluation.** Despite proposing a parameter-efficient finetuning method, there are no efficiency evaluations in the experimental results. Particularly since the approach adds parameters with respect to each task, how does that additional overhead trade off with accuracy? Aside from mentioning an improvement in number of parameters over a baseline in a single experimental setting, the efficiency compared to baselines is not discussed. Ideally the authors would report latency and/or memory requirements of the proposed approach compared to baselines to assess the practicality of the approach as the number of tasks increases. The parameter analysis experiments start to get at this, but they don’t even include the setting (BERT+AfriSenti) that benefitted most from the parameter expansion, for some reason. Differences in accuracy in these tables is small and could be due to random variation.
- **Missing related work.** The related work section was relegated to the appendix. In my opinion this is not appropriate and the paper should have been revised in order to properly make room for discussion of related work. Also, a lot of seemingly related work is missing, e.g. [Hyder et al. (2022)](https://arxiv.org/abs/2207.09074), [Wortsman, et al. (2020)](https://arxiv.org/abs/2006.14769), [Mirzadeh et al. (2020)](https://arxiv.org/abs/2010.04495), [Chaudhry et al. (2019)](https://arxiv.org/abs/1812.00420). Just some examples, not an exhaustive list.

**Questions:**

Questions:
- Why prefix tuning for BERT, AfroXLMR, and T5, and LoRA for Llama? Isn’t prefix tuning known to be hard to optimize, and therefore kind of bad? I would expect all PEFT approaches to be applied to all models, and especially LoRA to be applied to all models since it’s generally known to work better.
- How do SOTA and/or very basic methods for CL compare?
- Why was the Yelp dataset dropped from the sequence for Llama models? I understand that this was done in prior work, but I’d like to understand why, and whether the same reasoning is appropriate here.
- How does your method perform on longer task sequences, like the 15-task sequence from Wang et al. 2023 and Razdaibiedina et al. 2023?

Notes:
- The introduction is very long and takes a long time to get to the point where you describe your approach. To improve flow, I would recommend moving some of the discussion in the intro to background/related work sections and making the intro more concise so that a reader can more quickly get the gist of the proposed method and its motivation. It’s reasonable to assume that a reader is familiar with continual learning and catastrophic forgetting, for example, only a sentence or two is needed to establish that that is the focus area of the work in the intro.
- Fix references on lines 300-301, should be e.g. \citep not \citet.

---

> ### Author Response · Authors · 2024-11-28
>
> We thank the reviewer for their extensive and valuable feedback and wish to address their points and questions:
> - **Comparison to baseline CL methods & related work.** We have added a discussion of baseline CL methods such as MC-SGD and other methods mentioned by the reviewer to the related work section of the paper, and have added the related work centrally in the main body of the paper as suggested.
> Our focus in this work is on emphasizing knowledge transfer in true continual learning scenarios, where previous data is not available for replay. As such, we prioritize addressing existing gaps in methods designed for pre-trained language models, rather than including regularization- or replay-based approaches, which are known to provide only partial solutions to catastrophic forgetting [1,2].
> - **Evaluation metrics.** All evaluation metrics are first macro-averaged across random seeds and, if applicable, afterwards macro-averaged again over sequences, following our main prior work.
> As our approach and the presented baselines fully prevent catastrophic forgetting through their architectural design, backward transfer scores are identical to the reported forward transfer results and do not provide additional insights. Thus, we focus on a concise presentation of the results. Forward transfer is evaluated by comparing the final performance to a model only trained on one task, thus showcasing knowledge sharing potential. We understand that this reasoning was not made explicit in the paper and have added it to the experimental setup.
> - **Analysis of results & experimental setup.** We thank the reviewer for the suggestion of the MTL15 dataset. We have added an experiment with this dataset to the paper. Here, GE-PEFT also consistently outperforms existing state of the art approaches. Regarding the number of sequences, we have added an experimental extension to the appendix for results where improvements over baselines are small to show stability of GE-PEFT’s improvements.
> - **Efficiency evaluation.** As suggested, we have repeated the entire parameter efficiency study with the BERT+AfriSenti dataset and have reworked the entire parameter analysis section to better communicate the parameter efficiency of our proposed approach. We find that our approach is highly parameter efficient on strongly related tasks while less related tasks more rely on hyperparameter choices to achieve slow but meaningful growing.
>
> **Questions**
> - **Why prefix tuning for BERT, AfroXLMR, and T5, and LoRA for Llama?**
> Our main goal here was to show the feasibility of GE-PEFT on different PEFT methods, with prefix tuning still being a commonly used PEFT approach (e.g., [3,4]). While extensive explorations of the best combination of PEFT, language model, and GE-PEFT is a promising line for future work, we believe that the existing experimental setup is sufficient to show its feasibility and general applicability.
> - **How do SOTA and/or very basic methods for CL compare?** We evaluate MoCL and ProgPrompt as SOTA of CL, but focus on parameter-isolation methods in this work to guarantee the prevention of catastrophic forgetting.
> - **Why was the Yelp dataset dropped from the sequence for Llama models?** The exclusion of Yelp appears to be originating from the early work of [5] without explicit reason but has prevailed on many derivative works that constitute the SOTA in CL. We assume the exclusion was due to this dataset not coming with numbered classes instead of descriptive class names that are used by [5] for prompting generative language models.
> - **How does your method perform on longer task sequences?** We have added these experiments to the main paper, with GE-PEFT outperforming the current SOTA.
>
> [1] Wang, Liyuan, et al. "A comprehensive survey of continual learning: theory, method and application." IEEE Transactions on Pattern Analysis and Machine Intelligence (2024).
>
> [2] Razdaibiedina, Anastasia, et al. "Progressive Prompts: Continual Learning for Language Models." The Eleventh International Conference on Learning Representations.
>
> [3] Kim, Donghoon, et al. "Preserving Pre-trained Representation Space: On Effectiveness of Prefix-tuning for Large Multi-modal Models." Findings of the Association for Computational Linguistics: EMNLP 2024. 2024.
>
> [4] Zhang, Hongyi, et al. "Selective prefix tuning for pre-trained language models." Findings of the Association for Computational Linguistics ACL 2024. 2024.
>
> [5] Qin, Chengwei, and Shafiq Joty. "LFPT5: A Unified Framework for Lifelong Few-shot Language Learning Based on Prompt Tuning of T5." International Conference on Learning Representations.

---

### Meta-Review · Area_Chair_JxLL · 2024-12-20

**Metareview:**

All reviewers mentioned that the paper is well-written and the proposed method is simple to implement. However, I agree with the reviewers that the CL setting used for evaluations and baseline choice are not compelling. For example, in real-world scenarios the model is already trained in a wide variety of tasks and a couple of tasks need to be added to the model. Also none of the baselines are designed for CL scenarios (agree that EWC or a simple l2 distance on weights can provide a strong baseline). Moreover, an upper bound can be provided by training a model on mix of all tasks.
Finally, the reviewers bring up concerns about level of contributions of this work since it is mainly the combination of previously known methods.

**Additional Comments On Reviewer Discussion:**

None

---

### Decision · Program_Chairs · 2025-01-22

Reject